# The Role of Protein Arginine Methyltransferases in DNA Damage Response

**DOI:** 10.3390/ijms23179780

**Published:** 2022-08-29

**Authors:** Charles Brobbey, Liu Liu, Shasha Yin, Wenjian Gan

**Affiliations:** Department of Biochemistry and Molecular Biology, Hollings Cancer Center, Medical University of South Carolina, Charleston, SC 29425, USA

**Keywords:** arginine methylation, protein arginine methyltransferases, PRMT inhibitors, DNA damage

## Abstract

In response to DNA damage, cells have developed a sophisticated signaling pathway, consisting of DNA damage sensors, transducers, and effectors, to ensure efficient and proper repair of damaged DNA. During this process, posttranslational modifications (PTMs) are central events that modulate the recruitment, dissociation, and activation of DNA repair proteins at damage sites. Emerging evidence reveals that protein arginine methylation is one of the common PTMs and plays critical roles in DNA damage response. Protein arginine methyltransferases (PRMTs) either directly methylate DNA repair proteins or deposit methylation marks on histones to regulate their transcription, RNA splicing, protein stability, interaction with partners, enzymatic activities, and localization. In this review, we summarize the substrates and roles of each PRMTs in DNA damage response and discuss the synergistic anticancer effects of PRMTs and DNA damage pathway inhibitors, providing insight into the significance of arginine methylation in the maintenance of genome integrity and cancer therapies.

## 1. Introduction

Cells receive a constant onslaught on the genome from both endogenous and exogenous genotoxic stress, leading to numerous DNA lesions [1,2]. Among various types of DNA damage, including mismatches, base modifications, crosslinks, bulky adducts, and single/double-strand breaks (SSBs/DSBs), DSBs are the most serious and cytotoxic forms [3]. These damaged DNA can impede vital cellular processes including replication and transcription, which potentially causes cell cycle arrest and cell death if they are not repaired [4,5,6]. In mammalian cells, at least five major repair pathways are involved in repairing different types of DNA damage [7]. Mismatch repair (MMR) resolves single nucleotide mismatches generated during replication [8]. Base excision repair (BER) corrects covalent addition to DNA bases [9], while nucleotide excision repair (NER) clears bulky adducts and cross-linking lesions [10]. Homologous recombination (HR) and non-homologous end joining (NHEJ) are the two main repair pathways responsible for DSBs [11,12,13].

DNA repair is executed by a sophisticated cellular network, consisting of DNA damage sensors, transducers, and effectors, collectively termed DNA damage response (DDR) [14,15,16]. Three PI3K-related kinases, ATM, ATR, and DNA-PK, are positioned at the center of DDR [17]. The MRE11-RAD50-NBS1 (MRN) complex acts as a sensor of DSBs and contributes to the recruitment and activation of ATM [18]. ATRIP senses RPA-bound ssDNA and recruits ATR to DNA damage sites for activation [19]. The Ku70/80 heterodimer binds to DSBs and subsequently recruits and activates DNA-PK to promote NHEJ [20,21]. These active kinases phosphorylate and recruit hundreds of other transducers and effectors, such as H2AX, MDC1, 53BP1, BRCA1, TopBP1, Chk1, and Chk2, to complete the DNA repair process [17,22]. Aberrancy in the DDR protein network may lead to genome instability and is frequently associated with many human diseases, particularly cancers [23,24].

To ensure the efficient and proper repair of DNA damage, the DDR signaling is spatiotemporally regulated through multiple mechanisms [25]. Notably, various posttranslational modifications (PTMs) including phosphorylation, ubiquitylation, PARylation, methylation, acetylation, and sumoylation play prominent roles in controlling the dynamic recruitment and dissociation of DDR proteins at DNA breaks, [26,27]. The representative PTM event in DDR is the phosphorylation of histone H2AX at serine 139 (termed γH2AX) by ATM, ATR, and DNA-PK [28,29,30,31], which is a common marker of DNA damage and provides a platform for the assembly of downstream DDR proteins [32,33].

A wealth of studies in recent years has revealed that arginine methylation is as prevalent and versatile as serine phosphorylation and lysine ubiquitylation [34,35,36,37]. In mammals, a family of nine proteins called protein arginine methyltransferases (PRMTs) catalyzes the transfer of a methyl group from the donor S-adenosylmethionine (SAM) to the guanidino nitrogen atoms of arginine, which generates three forms of methylarginine: monomethylarginine (MMA), asymmetric dimethylarginine (ADMA), and symmetric dimethylarginine (SDMA). As such, these PRMTs can be categorized as three groups based on their catalytic activity: Type I includes PRMT1, PRMT2, PRMT3, CARM1 (Coactivator-associated arginine methyltransferase 1, also known as PRMT4), PRMT6, and PRMT8; Type II consists of PRMT5 and PRMT9; and Type III has PRMT7 only. Type I PRMTs register MMA and ADMA, while type II PRMTs generate MMA and SDMA. Type III catalyzes only MMA formation (Figure 1) [38,39]. Numerous downstream substrates of PRMTs have been identified, through which PRMTs regulate diverse fundamental processes, including transcription [40], RNA metabolism [41], DNA repair [42], signal transduction [43,44], and cell cycle control [45]. In this review, we will summarize the substrates of PRMTs and their roles in DDR.

## 2. Roles of PRMTs in DDR

### 2.1. Roles of PRMT1 in DDR

PRMT1 is the predominant methyltransferase that is responsible for approximately 90% of ADMA formation in mammalian cells [46,47]. Dysregulation of PRMT1 has been implicated in various human pathological conditions, such as breast cancer [48], prostate cancer [49], colon cancer [50], and other cancer types [51,52]. One of the critical roles of PRMT1 is maintenance of genome integrity. The mouse embryonic fibroblasts (MEFs) derived from *PRMT1* knockout mice display spontaneous DNA damage and polyploidy [53]. Several DDR proteins have been identified as PRMT1 substrates, including MRE11, BRCA1, 53BP1, Pol β, FEN1, APE1, and hnRNPUL1 (Figure 2).

#### 2.1.1. MRE11 (Meiotic Recombination 11)

MRE11, RAD50, and NBS1 form the MRN complex and function as the DSB sensor and transducer. After detecting the DNA damage signal, the MRN complex is directly recruited to the DSBs and initiates DNA end resection. Subsequently, it recruits ATM to the DNA damage sites for activation. In turn, activated ATM phosphorylates the MRN complex to further extend the DNA end resection for recruitment of other repair factors [54]. As the key component of the MRN complex, MRE11 possesses both endonuclease and exonuclease activities [55], which is essential for DNA end processing [56,57]. Aberrancy in MRE11 nuclease activity is associated with various human diseases, such as Alzheimer’s disease (AD) and cancers [58,59]. As such, extensive studies have focused on the regulatory mechanisms of MRE11, particularly PTMs [60], including arginine methylation as described below.

MRE11 contains a nuclease domain and a capping domain in the N-terminus, a DNA binding domain, and a glycine-arginine-rich (GAR) motif [61,62]. The arginine residues within the GAR motif are methylated by PRMT1, which is required for the exonuclease activity of MRE11, but the mechanism remains undefined. It is possible that methylation of the GAR motif causes MRE11 conformation change and thereby directly affects its enzyme activity or its binding to regulators. Moreover, arginine methylation of the GAR motif is required for the S phase checkpoint, but not for the MRN complex formation [63]. Further investigation found that mutating these arginine residues to lysine residues (MRE11^RK^, the non-methylable form of MRE11) leads to impairment in the recruitment of RPA and RAD51 to DSBs, ATR activation defects, and genomic instability. Interestingly, MRE11^RK^ MEFs display normal localization of MRN complex to the DSBs sites and ATM pathway activation in response to γ-irradiation (IR). It is possible that arginine methylation of MRE11 is recognized by certain readers, which specifically interacts with and activates ATR. As a result, MRE11^RK^ knock-in mice are hypersensitive to IR with all the mice dying within 2 weeks post-10 Gy of IR treatment [64]. A recent study found that GFI1 functions as an adaptor protein mediating interaction between PRMT1 and MRE11. Knockout of GFI1 results in a severe reduction in ADMA on MRE11, DNA repair defects, and cell death following DNA damage in T-cell acute lymphoblastic leukemia (T-ALL) [65]. However, GFI1 is mainly expressed in T lymphocytes. It remains to be determined how PRMT1-mediated methylation of MRE11 is regulated in cells without GFI1 expression.

#### 2.1.2. BRCA1 (Breast Cancer Type 1 Susceptibility Protein)

BRCA1 plays an important role in DSB repair by HR [66,67]. It is an 1836 amino acid protein and contains multiple functional domains interacting with a range of proteins [68]. The N-terminal RING domain binds to BARD1, which enhances the E3 ubiquitin ligase activity of BRCA1 [69,70]. The C-terminal BRCT domain is a reader of Ser/Thr phosphorylation and interacts with phosphorylated proteins, such as Abraxas and CtIP, which are associated with the recruitment of BRCA1 to DNA damage sites [67]. The coiled-coil domain in the middle region is required for BRCA1 interaction with PALB2 [71,72]. BRCA1 directly promotes HR repair by displacing the NHEJ protein 53BP1 from DSBs and enhancing end resection through interaction with CtIP, thereby recruiting downstream HR factors [73,74,75].

BRCA1 was reported to be methylated at the 504–802 region by PRMT1. Interestingly, arginine methylation of BRCA1 was significantly higher in breast cancer cells than in normal breast cells, suggesting it may affect the tumor suppressor function of BRCA1 [76]. A subsequent study showed that PRMT1-mediated methylation of BRCA1 is induced in response to IR. The silencing of PRMT1 likely prevents BRCA1 interaction with BARD1 and blocks the translocation of BRCA1 to the nucleus. As a result, the loss of PRMT1 impairs HR-mediated DSB repair [77]. However, the exact role of BRCA1 arginine methylation remains elusive, and the arginine methylation sites are yet to be defined.

#### 2.1.3. 53BP1 (p53-Binding Protein 1)

53BP1 is a key transducer/effector of the DDR and plays a crucial role in determining DSB repair pathway choice [78]. During the G1 phase of the cell cycle, 53BP1, together with RIF1 and PTIP, promotes NHEJ-mediated DSB repair and inhibits HR by blocking DNA end resection, while during the S/G2 phase, BRCA1 and CtIP remove 53BP1 to enable resection for HR repair [79]. 53BP1 loss restores HR repair and confers resistance to the PARP inhibitor in BRCA1-deficient cells [73,80,81].

53BP1 recruitment to DSB sites requires the minimal focus-forming region (FFR) composed of an oligomerization domain (OD), a GAR motif, a tandem Tudor domain, and a ubiquitylation-dependent recruitment (UDR) motif. The Tudor domain binds to dimethylated Lys20 of histone 4 (H4K20me2) and the UDR motif interacts with ubiquitylated H2AK15 (H2AK15ub) [82,83]. Studies have reported that the GAR motif is asymmetrically methylated by PRMT1, which is required for 53BP1 binding to single and double-stranded DNA. However, the mutation of arginine residues within the GAR motif does not affect 53BP1 foci formation upon topoisomerase II inhibitor treatment, implying methylation was not a prerequisite for 53BP1 recruitment to DSB sites. It was speculated that methylation of the GAR motif modulates the Tudor domain binding to histone or 53BP1 interaction with other DDR proteins to regulate the role of 53BP1 [84,85]. Like MRE11, GFI1 also mediates the 53BP1 interaction with PRMT1 to promote its arginine methylation in GAR [65]. The identification of GAR motif binding proteins or readers of methylated GAR would be helpful to dissect the exact role of PRMT1-mediated methylation of 53BP1.

#### 2.1.4. Pol β (DNA Polymerase β)

Pol β is the major polymerase in the BER pathway that remove apurinic/apyrimidinic (AP) sites from DNA [86]. It is involved in two steps of BER: 5′-end deoxyribose phosphate (dRP) removal and gap-filling DNA synthesis, which is carried out by the N-terminal lyase domain and the C-terminal polymerase domain [87]. A recent study also showed that Pol β is involved in alternative NHEJ (aNHEJ) repair [88].

Available data show that PRMT1 directly interacts with Pol β via its lyase domain and monomethylates it at arginine 137. This methylation does not affect Pol β lyase and polymerase activity but disrupts its interaction with PCNA [89]. Given that PCNA enhances Pol β-dependent long-patch BER (LP-BER) [90], it is possible that the PRMT1-mediated methylation of Pol β serves as a termination signal to inactivate Pol β when the repair is complete.

#### 2.1.5. FEN1 (Flap Endonuclease 1)

FEN1 is a structure-specific nuclease that participates in various DNA repair pathways, including LP-BER, MMR, NER, and HR. It recognizes the single-strand flap and cleaves it to create a nick, which is filled by DNA ligase1 [91,92]. Overexpression of FEN1 is observed in many cancer types and is correlated with cancer aggressiveness [93]. The activity and DNA binding ability of FEN1 are regulated by several PTMs, including phosphorylation, acetylation, and methylation [94].

PRMT1 catalyzes ADMA on FEN1 and thereby enhances FEN1 protein stability, but does not alter FEN1 mRNA level, localization, and its interaction with PCNA. PRMT1 knockdown decreases FEN1 expression and increases DNA damage in A549 lung cancer cells treated with Temozolomide (TMZ) or 5-fluoro-uracil, which can be partially rescued by overexpressing FEN1 [95]. These results suggest that FEN1 is an important downstream substrate of PRMT1 in DDR. However, the methylated arginine site(s) has not yet been identified, preventing evaluation of the physiological significance of PRMT1-mediated arginine methylation in FEN1-dependent DNA repair.

#### 2.1.6. APE1 (Apurinic/Apyrimidinic Endonuclease 1)

APE1 is a multifunctional and ubiquitous protein responsible for most endonuclease activities. It also possesses exonuclease and RNA cleavage capabilities [96,97]. APE1 is a key component of the BER pathway. It cleaves the 5′ phosphodiester bond to generate dRP for further processing of the AP sites by Pol β [98,99].

PTMs, including acetylation, phosphorylation, ubiquitination, and methylation, play crucial roles in regulating APE1 activities, localization, and protein stability [100,101]. A recent study found that APE1 is methylated at arginine 301 (R301) by PRMT1, which is enhanced by oxidative agents, such as H_2_O_2_ and menadione [102]. This methylation does not affect its nuclease activity but enhances APE1 binding to the mitochondrial outer membrane translocase, TOM20, and thus promotes APE1 translocation to the mitochondrion. Deficiency in R301 methylation increases DNA damage in mitochondria and sensitizes cells to oxidative stress. Further studies are warranted to define the role of PRMT1-mediated methylation in oxidative DNA damage. For example, identification of PRMT1 as a target of oxidative agents using label-free methods [103] and genome-wide mapping of PRMT1-associated DNA lesions induced by oxidants using DNA–protein cross-linking sequencing [104]

#### 2.1.7. hnRNPUL1 (Heterogeneous Nuclear Ribonucleoprotein U-like Protein 1)

hnRNPUL1, also known as adenovirus early region 1B-associated proteins 5 (E1B-AP5), is a member of the heterogenous nuclear ribonucleoprotein family that mainly functions in RNA splicing, stabilization, decay, and transcription [105].

In addition to RNA metabolism, hnRNPUL1 has been shown to play a role in DDR [106]. It interacts with the MRN complex via NBS1 and is recruited to DSB sites in the presence of transcription inhibitors, which is dependent on the GAR motif of hnRNPUL1. Subsequently, hnRNPUL1 promotes DNA end resection and ATR-dependent signaling, leading to the promotion of HR. As a result, hnRNPUL1 depletion renders cells sensitive to DSB-inducing agents. To further elucidate the role of the GAR motif in hnRNPUL1-mediated DNA repair, Gurunathan et al. showed that PRMT1 interacts with hnRNPUL1 and methylates R612, R618, R620, R639, R645, R656, and R661 that are located at the GAR motif. PRMT1 knockdown or mutating these arginine sites to lysine blocks the methylation signal and reduces its interaction with NBS1 and recruitment to DSB sites upon treatment of transcription inhibitors [107]. Therefore, it was proposed that the roles of hnRNPUL1 in RNA metabolism and DSB repair are likely determined by PRMT1-mediated methylation at the GAR motif.

### 2.2. Roles of PRMT2 in DDR

PRMT2 was first identified by its sequence homology to PRMT1 in the human genome with 27% identity and 50% similarity [108,109]. Although PRMT2 contains the canonical catalytic domain as other PRMTs, it exhibits low methyltransferase activity towards histones H3 and H4 [110], and undergoes strong automethylation [111]. No developmental defects and malignancies are found in *PRMT2* knockout mice [112]. PRMT2 expression is dysregulated in many cancers, such as breast cancer and hepatocellular carcinoma [113,114].

RNA-seq and pathway analysis showed that multiple genes involved in DNA damage are downregulated and the DNA repair pathway is suppressed in PRMT2-knockdown MCF-7 cells, suggesting a critical role of PRMT2 in DDR [115]. Indeed, PRMT2 knockdown promotes the clearance of cyclobutane pyrimidine dimers (CPDs) after ultraviolet (UV) radiation, indicating the role of PRMT2 in BER or NER. Moreover, using the DR-GFP recombination reporter system, it was shown that PRMT2 depletion increases DSB repair through HR. However, BRCA1, a known positive regulator of HR, is decreased upon PRMT2 depletion. It is unclear how PRMT2 depletion promotes HR repair with BRCA1 downregulation. Further studies are needed to fully dissect the role of PRMT2 in DDR, including identifying DDR proteins as PRMT2 substrates.

### 2.3. Roles of CARM1 (PRMT4) in DDR

CARM1 is a type I PRMT that methylates R17/R26 of histone H3 [116,117] and non-histone substrates, such as BAF155 [118]. CARM1 regulates diverse cellular processes including transcription, RNA splicing, and cell cycle to control autophagy, metabolism, and development [119]. Of note, embryos in CARM1 knockout mice survived the course of development but died right before birth (E19.5), suggesting that CARM1 is an essential gene for organism growth [120]. Like other PRMTs, CARM1 dysregulation has been implicated in colorectal cancer, leukemia, breast, and prostate cancer [121].

Regarding the role of CARM1 in DDR, it has been evidenced that CARM1 dimethylated p300 at R754 promotes p300 interaction with BRCA1, leading to enhancement of the BRCA1 transcriptional activity (Figure 3). CARM1 knockdown suppresses DNA damage-induced expression of p21 and GADD45, two important regulators of cell cycle arrest. Importantly, overexpression of p300 wild-type, but not the p300 R754A mutant, enhances p21 expression in response to DNA damage, suggesting the significance of R754 methylation. Mechanistically, CARM1 and p300 promote BRCA1 recruitment to the p21 promoter and induce its transcription [122]. Even though CARM1 may indirectly modulate DNA repair through p21, it will give a clear vista if DDR proteins can be identified as its direct substrates.

### 2.4. Role of PRMT5 in DDR

As the predominant Type II PRMT, PRMT5 is distributed in both cytoplasm and nucleus, where it methylates histones and non-histone substrates to exert versatile functions [123,124]. PRMT5 forms an octameric complex with MEP50 to optimize its methyltransferase activity [125]. Its overexpression has been documented in several different cancer types and has largely been tagged as an oncoprotein [126]. Consequently, PRMT5 has gained tremendous interest as a potential antitumor target and dozens of inhibitors have been developed with some being evaluated in clinical trials. Numerous PRMT5 substrates have been identified, which are involved in transcription, RNA splicing, and signal transduction [44]. It is not surprising that PRMT5 also serves as a crucial regulator of genome stability through methylating DDR proteins, including 53BP1, FEN1, RAD9, RUVBL1, and TDP1 (Figure 4).

#### 2.4.1. 53BP1 (p53-Binding Protein 1)

In addition to PRMT1-mediated ADMA, a recent study by Hwang et al. showed that 53BP1 also undergoes SDMA on the GAR motif by PRMT5 [127]. As such, PRMT5 knockdown decreases SDMA but increases ADMA of 53BP1, whereas PRMT1 depletion reverses this observation, indicating that PRMT5 and PRMT1 compete to methylate 53BP1 on the GAR motif. Notably, either PRMT5 depletion or mutation of the five arginine residues to lysine residues (5RK) within the GAR motif shortens the half-life of 53BP1 protein, suggesting that PRMT5-mediated SDMA increases 53BP1 protein stability. Consequently, compared to control cells, PRMT5-depleted cells display a reduction in the overall intensity and the number of 53BP1 foci, leading to impairment of NHEJ and HR-mediated repair process and sustained γH2AX levels. Importantly, these repair defects were rescued by overexpression of 53BP1, suggesting PRMT5 exerts its functions in DDR in part through the stabilization of the 53BP1 protein. These studies provide a fascinating example whereby ADMA and SDMA coordinatively finetune DDR to control DNA repair. However, the mechanisms underlying how PRMT5 controls 53BP1 turnover remain to be explored. Given the E3 ligases RNF8/RNF168/RNF146-mediated ubiquitination and NUDT16-mediated removal of ADP-ribosylation have been involved in 53BP1 protein degradation [128,129], it would be interesting to explore crosstalk among these mechanisms.

#### 2.4.2. FEN1 (Flap Endonuclease 1)

Like 53BP1, FEN1 is another DDR protein that is co-regulated by both PRMT1 and PRMT5. It interacts with PRMT5 and is symmetrically dimethylated at four arginine residues (R19, R100, R104, and R192) with R192 as the primary methylation site [130]. This methylation event leads to a decrease in cyclin E/CDK2-mediated phosphorylation of FEN1 at S187, thereby enhancing its interaction with PCNA to promote LP-BER. Cells expressing the methylation deficient FEN1 mutant (R192K or 4RK) display higher γH2AX levels compared to cells expressing FEN1 wild-type, indicating accumulation of DSBs. Consequently, these cells are more sensitive to oxidative stresses. It is proposed that FEN1 arginine methylation promotes its loading on flap sites via interaction with PCNA and then offload Pol β or Pol δ, allowing FEN1 to remove flap structure. After cleavage, FEN1 is demethylated and re-phosphorylated by cyclin E/CDK2, following dissociation from DNA. A subsequent study found that cyclin E/CDK2-mediated phosphorylation of FEN1 at S187 is required for SUMOlyation and cell cycle-dependent proteasomal degradation of FEN1 [131]. Thus, it is speculated that like PRMT1, PRMT5 also plays a role in the modulation of FEN1 protein stability.

#### 2.4.3. RAD9

RAD9 is evolutionarily conserved from yeast to human and has multiple functions, including cell cycle checkpoint, DNA repair, radioresistance, and apoptosis [132]. The N-terminal part of RAD9 is responsible for interaction with HUS1 and RAD1, thus forming a heterotrimer called a 9-1-1 complex [133]. The C-terminal portion harbors a nuclear localization signal and is phosphorylated by multiple kinases, which is required for RAD9 translocation to the nucleus and its interaction with other DDR proteins including RPA and TopBP1 [134]. In response to DNA damage, the 9-1-1 complex is recruited to the damaged sites with the assistance of the RAD17-RFC2-5 complex and then activates the ATR-mediated signaling pathway and plays a role in HR, BER, NER, MMR, and alternative NHEJ (aNHEJ) [134,135,136]. For example, RAD9 influences HR by interacting with RAD51 and affects BER through stimulating enzymatic activities of APE1 and Pol β [137].

PRMT5 directly binds and methylates RAD9 in vitro and in cells [138]. Mutation of three arginine sites, R172, R174, and R175, to lysine or alanine (3RK or 3RA) or knockdown of PRMT5, abolished the methylation signal, suggesting that these three arginine residues are the major methylation sites by PRMT5. Of note, this methylation event does not affect the 9-1-1 complex formation but is induced by DNA damaging agent, hydroxyurea (HU). Cells expressing RAD9 3RA or 3RK are defective in the G2/M checkpoint and ATR-dependent Chk1 phosphorylation in response to HU or IR treatment. However, these cells are more sensitive to HU, but not IR, indicating PRMT5-mediated methylation of RAD9 may be required for SSB repair but dispensable for DSB repair.

#### 2.4.4. RUVBL1 (RuvB-like 1)

RUVBL1 and its homolog RUVBL2 are AAA+ ATPases and are involved in chromatin remodeling, transcription, and DNA repair. They act as a scaffolding protein to participate in diverse protein complexes, such as INO80, SWR1, YY1, and TIP60, all of which have been implicated in DDR [139,140]. The RUVBL1-containing TIP60 acetyltransferase complex is recruited to DNA damage sites and is required for acetylation of histone H4 or H2AX, which is critical for the dephosphorylation of γH2AX [141,142]. In addition, the TIP60 complex-mediated acetylation of histone H4K16 impairs 53BP1 binding to methylated H4K20, and consequently increases BRCA1 localization to DSBs to promote HR [143]. RUVBL1/2 also promotes RAD51 foci formation during the HR repair process [144,145].

In demystifying the mechanism associated with PRMT5 in DDR, RUBVL1 was identified as a PRMT5 interacting partner and substrate that is symmetrically dimethylated on R205 [146]. Depletion of RUBVL1 impaired HR-mediated DSB repair with increased γH2AX and 53BP1 foci upon IR exposure. The reintroduction of RUBVL1 wild-type, but not the methylation deficient RUBVL1 R205K mutant, rescued these repair defects. Notably, RUBVL1 methylation promotes TIP60-mediated acetylation of histone H4K16, leading to the dissociation of 53BP1 from DSBs. Interestingly, both RUBVL1 wild-type and the R205K mutant were effectively recruited to DNA damage sites, suggesting that arginine methylation does not regulate TIP60 chromatin association. This study provides another example that PRMT5 regulates the choice of NHEJ and HR-dependent DSB repair.

#### 2.4.5. TDP1 (Tyrosyl-DNA Phosphodiesterase 1)

TDP1 is an important DNA repair enzyme for the removal of trapped Top I cleavage complexes (TopIcc) that are generated by either DNA lesions (mismatches, abasic sites, nicks, and adducts) or Top I inhibitor (Camptothecin, CPT) during Top I-mediated cleavage–religation process. TopIcc may stall the progression of replication and transcription forks and generate DSBs. TDP1 interacts with other DDR proteins including PARP1, Ligases III, XRCC1, and PNKP, and has been engaged in BER, NHEJ, and HR [147,148,149,150].

TDP1 consists of two domains: the N-terminal domain that is critical for its protein stability and recruitment to DNA damage sites and the C-terminal catalytic domain that hydrolyzes the phosphodiester bond between Top I tyrosyl moiety and the DNA 3′-end [151]. PRMT5 interacts with the N-terminus of TDP1 and symmetrically dimethylates it at R361 and R586, which is enhanced by CPT treatment in a DNA replication-dependent manner [152]. Notably, PRMT5-mediated arginine methylation promotes TDP1 catalytic activity and is required for TDP1 interaction with XRCC1. This methylation event is critical for the repair of CPT-induced TopIcc and protects cells against CPT treatment.

### 2.5. Role of PRMT6 in DDR

PRMT6 catalyzes MMA and ADMA on multiple sites of histones, including H3R2, H3R17, H3R42, and H2AR29, through which it possesses both transcriptional repression and activation roles [153,154,155,156,157]. PRMT6 promotes H3R2me2a to antagonize the activating mark H3K4me3, leading to the suppression of tumor suppressors expression, such as p53 and p21 [156,158]. In contrast, PRMT6 generates H3R42me2a to stimulate p53-dependent transcription [153]. It is unclear how these arginine modifications of histones by PRMT6 coordinatively control transcription. PRMT6 also regulates several non-histone substrates that are involved in cancers and cardiac diseases [159,160,161].

Like PRMT1, PRMT6 also binds the lyase domain of Pol β to promote methylation of Pol β at R83 and R152, which increases in response to methyl methane sulfonate (MMS) treatment. Functionally, R83/R152 methylation enhances Pol β polymerase activity, DNA binding, and processivity to promote LP-BER (Figure 5) [162]. Therefore, PRMT1 and PRMT6 regulate LP-BER by catalyzing the methylation of Pol β at different sites for modulation of Pol β functions through different mechanisms. These modifications likely offer flexibility to tightly control Pol β for DNA repair in different tissues or cell types.

PRMT6 is also functionally redundant with CARM1 [155]. Both CARM1 and PRMT6 can deposit the H3R17me2a mark. CARM1/PRMT6 double knockout mouse embryonic fibroblasts (MEFs) display higher basal γH2AX staining than single knockout MEFs. Consistently, a combination of CARM1 and PRMT6 inhibitors enhances γH2AX levels and has synergistic effects on inhibitors on cell proliferation. However, it remains unknown whether CARM1 and PRMT6 share common DDR proteins as substrates or act on different repair pathways to maintain genome integrity.

### 2.6. Role of PRMT7 in DDR

Unlike other PRMTs, PRMT7 catalyzes only the MMA formation and contains two putative SAM binding motifs, both of which are necessary for its methyltransferase activity [163,164]. PRMT7 methylates histone H2B at R29/R31/R33 and histone H4 at R17/R19 [164]. Non-histone substrate of PRMT7 include Dvl3 [165], G3BP2 [166], eIF2α [167], and Hsp70 [168]. PRMT7 overexpression has been implicated in breast cancer and leukemia [169,170].

PRMT7 also plays a role in DDR. It is enriched on the promoter region of DNA repair genes, including *ALKBH5*, *APEX2*, *POLD1*, and *POLD2*, and methylates the H2AR3 and H4R3, leading to the transcriptional repression of these genes (Figure 6). As a result, PRMT7 depletion confers resistance to cisplatin, chlorambucil, and mitomycin C, but not doxorubicin. Interestingly, depletion of POLD1, but not ALKBH5, APEX2, or POLD2, re-sensitizes PRMT7-depleted cells to these DNA damaging agents [171]. In contrast, PRMT7 knockdown enhances sensitivity to CPT, indicating that it may play a role in TopIcc and BER [172]. Hence, PRMT7 regulation of DNA damage appears to be cell type or genotoxin specific. Identifying DDR substrates of PRMT7 would reveal how PRMT7 specifically modulates DNA repair and sensitivity to DNA damage agents.

### 2.7. Role of PRMT8 in DDR

Compared to other PRMTs, PRMT8 displays some unique characteristics: it is largely expressed in brain and neurons [173]; it is localized to the plasma membrane by myristoylation at the Gly2 residue, while the N-terminus (1–60 aa) suppresses its catalytic activity [174]; and it also possesses phospholipase activity that directly hydrolyzes phosphatidylcholine (PC) [175]. Several specific substrates of PRMT8 have been identified, including NIFK [176] and voltage-gated sodium channel Nav1.2 [177]. PRMT8 has been implicated in neurological diseases [178].

Even though PRMT8 substrates in DDR have not yet been identified, available evidence indirectly shows their relevance in DNA repair. Simandi et al. showed that motor neurons (MNs) from aged PRMT8 knockout mice exhibit increased γH2AX levels, indicating accumulation of unrepaired DNA. Consistently, pathway analysis of differentially expressed genes in control and PRMT8 knockout spinal cord samples identifies DNA replication, recombination, and repair as some of the top affected pathways [179].

### 2.8. Role of PRMT3 and PRMT9 in DDR

PRMT3 is a distinct Type I PRMT, which displays different subcellular localization and substrate specificity [180,181]. Several substrates of PRMT3 have been identified, such as the 40S ribosomal protein S2 (RpS2) [182,183], the nuclear poly(A)-binding protein (PABPN1) [184], and others [185,186]. Little is known about PRMT9 as spliceosome-associated protein 145 (SAP145, also known as SF3B2) is its sole substrate to date [187,188]. The functions of PRMT3 and PRMT9 in DDR have not yet been studied.

## 3. Synergistic Effects of PRMT and DDR

Due to the toxicity and increased resistance to monotherapy during cancer treatment, extensive studies have focused on identifying novel targets or strategies that display synergistic effects for overcoming these drawbacks to improve therapeutic outcomes. Numerous combinatorial treatments have been approved by The United States Food and Drug Administration (FDA) or are currently being evaluated in clinical trials for various cancers [189,190]. For example, the combination of the PARP inhibitor (Olaparib) and anti-angiogenesis agent (Bevacizumab) was approved by FDA in 2020 as the first-line maintenance treatment of ovarian cancer patients with HR deficiency [191,192].

Multiple PRMT inhibitors have been developed and are undergoing evaluation in clinical trials (Table 1) [193,194].

However, the phase I trial of type I PRMT inhibitor GSK3368715 has been terminated in part due to serious adverse events. Several defects of PRMT5 inhibitors have also emerged in preclinical studies, such as resistance in triple-negative breast cancer cells [195] and suppression of immune response [196,197,198]. Since PRMTs are crucial regulators of DDR, targeting DDR can be a potential strategy for enhancing the anti-tumor efficacy of PRMT inhibitors to achieve better outcomes (Table 2).

Dominici et al. screened a small molecule library of epigenetic and anticancer drugs and identified PARP inhibitors as the top synergistic compounds with the type I PRMT inhibitor MS023 in A549, a human *MTAP*-negative non-small cell lung carcinoma (NSCLC) cell line. Compared to mono-treatment, co-treatment of A549 cells with low doses of MS023 and the PARP inhibitor BMN-677 significantly increases cell death and γH2AX levels, indicating that the elevated cytotoxicity is in part due to aggravated DNA damage. Notably, MS203 treatment overcomes PARP inhibitor resistance [199]. Similarly, a recent study showed that PRMT5 loss impairs HR-mediated DNA repair in Leukemia cells through aberrant splicing of TIP60. As a result, PRMT5 inhibitor GSK3186000A renders hemopoietic cells vulnerable to PARP inhibitor Olaparib. Importantly, the synergism is also observed in leukemic cells resistant to PRMT5 inhibitor [200]. Hence, the combination of PRMT and PARP inhibitors is a potential option for combating HR-proficient cancers.

PRMT inhibition has also been shown to enhance sensitivity to DNA damaging agents. In ovarian cancer cells, PRMT1 promotes the expression of genes involved in senescence-associated secretory phenotype (SASP) in response to cisplatin (CCDP), which is believed to halt the cell cycle and protect cells from death. As result, genetic depletion of PRMT1 or the type I PRMT inhibitor MS023 increases the cytotoxicity of cisplatin [201]. Likewise, depletion of PRMT1 impairs DNA repair and increases apoptosis in response to etoposide or 5-fluorouracil (5-FU) or temozolomide (TMZ) [53,95]. Recent studies have also documented that the combination of PRTM5 inhibition and gemcitabine synergistically decreases tumor growth in a patient-derived pancreatic cancer xenograft model in part due to excessive unrepaired DNA damage [202], while knockdown of PRMT5 augments the antiproliferative effect of etoposide in breast cancer cells [127].

## 4. Conclusions and Perspectives

As a common PTM, arginine methylation regulates DDR signaling by modulating DDR protein transcription, RNA splicing, protein stability, binding partners, enzymatic activities, and localization. Intriguingly, a DDR protein can be methylated by multiple PRMTs, leading to diverse outcomes. Moreover, PRMT-mediated regulation of DDR proteins appears to be cell-type and genotoxin specific. These unveil the complexity of arginine methylation in DDR.

In the past decade, the expanded list of arginine-methylated DDR proteins further emphasizes the significance of PRMTs in DNA damage repair. However, the research on this topic is still at an early stage and plenty of questions remain to be answered. What are the upstream regulators that control PRMTs activity upon DNA damage and their recruitment to DNA damage sites? It has been reported that DNA-PK-dependent phosphorylation of PRMT1 is required for its recruitment to chromatin in response to replication stress [201], while Src kinase phosphorylates PRMT5 to impede its enzymatic function in NHEJ repair [127]. If and how do other PTMs, such as acetylation, ubiquitination, and methylation, regulate PRMTs in DDR? Moreover, what are the readers of arginine-methylated DDR proteins that may serve as a platform for the assembly of DNA repair complexes? How is the arginine methylation signal terminated when the DNA repair is completed? Does arginine methylation interplay with other PTMs, such as phosphorylation and ubiquitination, to coordinatively regulate the DDR signaling pathway? Addressing these questions may significantly advance our knowledge on arginine methylation-mediated regulation of genome stability and identify novel targets and strategies to combat cancers.

## Figures and Tables

**Figure 1 ijms-23-09780-f001:**
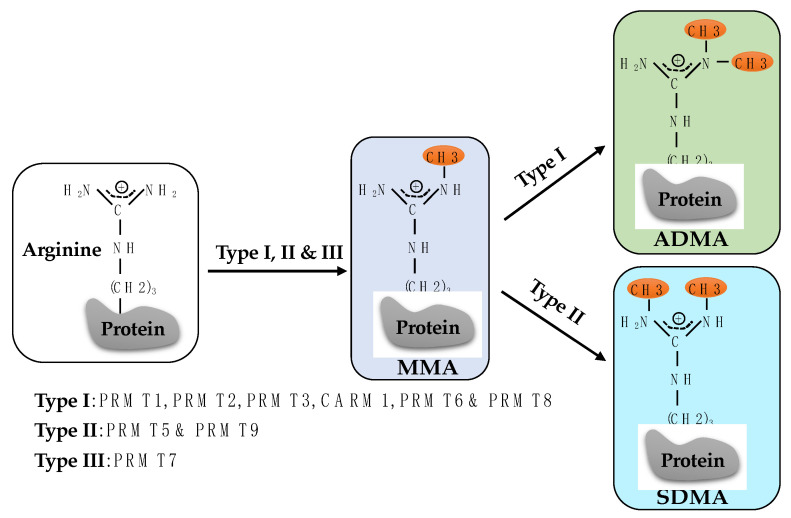
Three forms of protein arginine methylation, including MMA, ADMA, and SDMA, are catalyzed by nine PRMTs (Type I, II, and III). MMA, monomethylarginine; ADMA, asymmetric dimethylarginine; SDMA, symmetric dimethylarginine.

**Figure 2 ijms-23-09780-f002:**
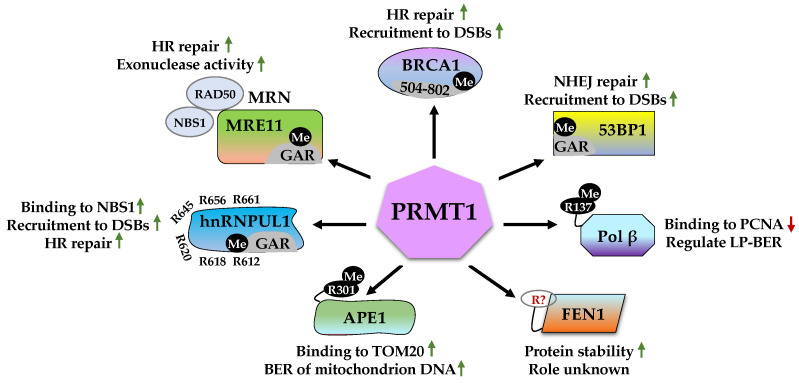
PRMT1 functions as a critical DDR regulator by methylating MRE11, BRCA1, 53BP1, Pol β, FEN1, APE1, and hnRNPUL1. MRE11, Meiotic recombination 11; BRCA1, Breast cancer type 1 susceptibility protein; 53BP1, p53-binding protein 1; Pol β, DNA polymerase β; FEN1, Flap endonuclease 1; APE1, Apurinic/apyrimidinic endonuclease 1; hnRNPUL1, heterogeneous nuclear ribonucleoprotein U-like protein 1; GAR, glycine-arginine-rich motif; NBS1, Nijmegen breakage syndrome 1; DSBs, DNA double-strand breaks; HR, homologous recombination; NHEJ, non-homologous end joining; PCNA, proliferating cell nuclear antigen; TOM20, translocase of outer mitochondrial membrane 20; BER, base excision repair; LP-BER, long-patch BER; Me, methylation. Green arrows represent “increase” and red arrows represent “decrease”.

**Figure 3 ijms-23-09780-f003:**
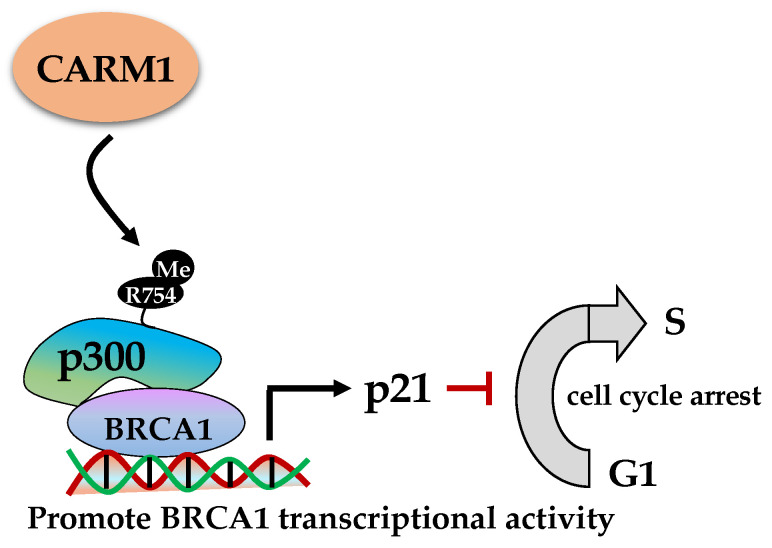
CARM1 methylates p300 to enhance BRCA1 transcriptional activity towards p21, leading to G1 cell cycle arrest upon DNA damage. BRCA1, Breast cancer type 1 susceptibility protein.

**Figure 4 ijms-23-09780-f004:**
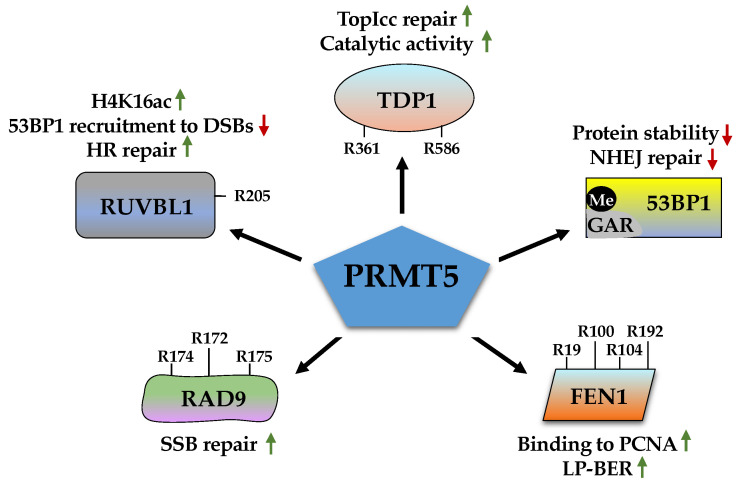
PRMT5 regulates different DNA repair pathways by methylating various DDR proteins, including 53BP1, FEN1, RAD9, RUVBL1, and TDP1. 53BP1, p53-binding protein 1; FEN1, Flap endonuclease 1; RUVBL1, RuvB-like 1; TDP1, Tyrosyl-DNA phosphodiesterase 1; TopIcc, Top I cleavage complexes. Green arrows represent “increase and red arrows represent “decrease”.

**Figure 5 ijms-23-09780-f005:**
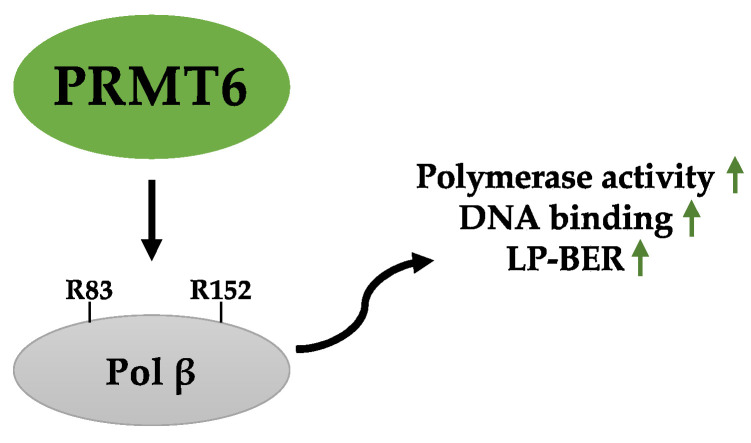
PRMT6-mediated methylation of Pol β enhances its activity and binding of DNA to promote LP-BER. Pol β, DNA polymerase β. Green arrows represent “increase”.

**Figure 6 ijms-23-09780-f006:**
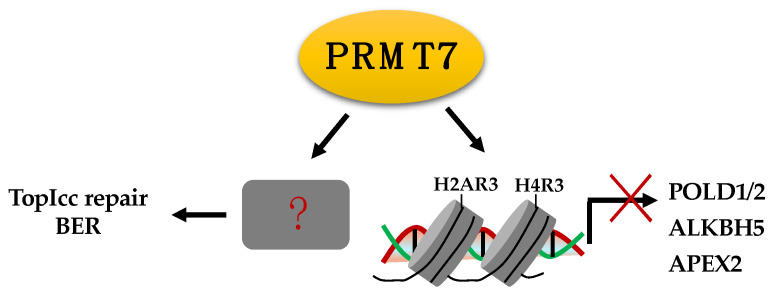
PRMT7 suppresses transcription of DNA repair genes by methylating H2AR3 and H4R3, and it is likely involved in TopIcc repair and BER via unknown mechanisms. ALKBH5, AlkB homolog 5; APEX2, Apurinic/apyrimidinic endodeoxyribonuclease; POLD1, DNA polymerase delta 1; POLD2, DNA polymerase delta subunit 2.

**Table 1 ijms-23-09780-t001:** PRMT inhibitors in clinical trials.

Drug Name	Targeted PRMTs	Trial Number	Disease or Condition	Status
GSK3368715	Type I PRMTs	NCT03666988	DLBCL and MTAP-deficient solid tumors	Terminated
GSK3326595	PRMT5	NCT04676516	Breast cancer	Not yet recruiting
JNJ-64619178	PRMT5	NCT03573310	Solid tumor, Non-Hodgkin Lymphoma, myelodysplastic syndromes	Active, not recruiting
PF-06939999	PRMT5	NCT03854227	Metastatic NSCLC, HNSCC, esophageal cancer, endometrial cancer, cervical cancer, and bladder cancer	Terminated
PRT811	PRMT5	NCT04089449	Advanced solid tumors, CNS lymphoma, and glioma	Recruiting
PRT543	PRMT5	NCT03886831	Advanced solid tumors and hematologic malignancies	Active, not recruiting
TNG908	PRMT5	NCT05275478	Patients with MTAP-deleted advanced or metastatic solid tumors	Recruiting
MRTX1719	PRTM5-MTA	NCT05245500	Patients with MTAP-deleted advanced or metastatic solid tumors	Recruiting

**Table 2 ijms-23-09780-t002:** Inhibition of PRMT1 or PRMT5 synergizes with PARP inhibitors or DNA damaging agents.

Inhibition of PRMT1 or PRMT5	Synergistic Agents	Cancer Type
Type I PRMT inhibitor (MS023)	PARP inhibitor (BMN-677)	Lung Cancer [199]
Type I PRMT inhibitor (MS023)	Cisplatin	Ovarian cancer [201]
Knockdown of PRMT1	Etoposide	Osteosarcoma [53]
Knockdown of PRMT1	TMZ or 5-FU	Lung Cancer [95]
PRMT5 inhibitor (EPZ015666)	Gemcitabine	Pancreatic cancer [53]
Knockdown of PRMT5	Etoposide	Breast cancer [127]
PRMT5 inhibitor (GSK3186000A)	PARP inhibitor (Olaparib)	AML [200]

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
