# Peer review of "The Role of Protein Arginine Methyltransferases in DNA Damage Response"

_ijms, 2022, doi:10.3390/ijms23179780_

Round 1
Reviewer 1 Report
The manuscript ‘The role of protein arginine methyltransferases in DNA damage response’ by Brobbey et al. adds good information in the field of methyltransferases and their importance in genomic integrity, and cancer research. Authors have reviewed different classes of arginine methyltransferases, and their roles during the DNA damage response. Indeed, the review sheds light on PRMTs, their substrates, functional relevance in DDR, and targeting PRMTs by inhibitors.
The specific comments, which could help to improve the manuscript, are:
1. Line 57: PRTMs should be replaced with PRMTs.
2. Line 59: Three forms of chemical modifications MMA, ADMA, and SDMA can be shown pictorially to understand better.
3. Line 74: Dysregulation of PRMT1 has been implicated in cancer. It will be better to mention which type of cancer.
4. Figure 1: hnRNPUL1 box doesn’t mention MeGAR, however in the text, it has been described.
5. Line 83: MRN complex is MRE11, Rad50, and NBS1. Hence, reorder their names in this line.
6. Line 107: Mre11RK should be replaced with MRE11RK
7. While describing all PRMTs, PRMT3 and PRMT9 were not touched. What is the reason? Is there not enough literature? If yes, mention it in the review.
8. Line 232: CARM1 can be replaced with CARM1 (PRMT4).
Author Response
The manuscript ‘The role of protein arginine methyltransferases in DNA damage response’ by Brobbey et al. adds good information in the field of methyltransferases and their importance in genomic integrity, and cancer research. Authors have reviewed different classes of arginine methyltransferases, and their roles during the DNA damage response. Indeed, the review sheds light on PRMTs, their substrates, functional relevance in DDR, and targeting PRMTs by inhibitors.
Response: We thank the reviewer very much for recognizing the significance of our work.
The specific comments, which could help to improve the manuscript, are:
1. Line 57: PRTMs should be replaced with PRMTs.
Response: We thank the Reviewer for picking up this typo, which has been corrected.
2. Line 59: Three forms of chemical modifications MMA, ADMA, and SDMA can be shown pictorially to understand better.
Response: We thank the reviewer very much for this excellent suggestion. These forms of arginine methylation have been shown pictorially in revised Figure 1.
3.Line 74: Dysregulation of PRMT1 has been implicated in cancer. It will be better to mention which type of cancer.
Response: We thank the reviewer for this excellent suggestion. The cancer types have been included.
4. Figure 1: hnRNPUL1 box doesn’t mention MeGAR, however in the text, it has been described.
Response: We thank the reviewer for pointing out this. The Figure has been modified to include MeGAR in the hnRNPUL1 box.
5. Line 83: MRN complex is MRE11, Rad50, and NBS1. Hence, reorder their names in this line.
Response: We thank the reviewer for pointing out this. It has been reordered.
6. Line 107: Mre11RK should be replaced with MRE11RK
Response: We thank the reviewer for pointing out this, which has been corrected.
7. While describing all PRMTs, PRMT3 and PRMT9 were not touched. What is the reason? Is there not enough literature? If yes, mention it in the review.
Response: We thank the reviewer for raising this concern. No study has reported the role of PRMT3 and PRMT9 in DDR. We have provided the explanation in the section 2.8.
8. Line 232: CARM1 can be replaced with CARM1 (PRMT4).
Response: We thank the reviewer for this suggestion. CARM1 has been replaced with CARM1 (PRMT4).
Reviewer 2 Report
The authors present a review the current understanding of the protein arginine methyltransferases in relation to the DNA damage response pathways. This is the first major review of this family of proteins in more than 6 years. I have no criticisms of the subject or scope. The authors have incorporated much of what is currently known making this manuscript a useful reference. Minor point, there is a typo on line #290: cycline → cyclin.
Author Response
The authors present a review the current understanding of the protein arginine methyltransferases in relation to the DNA damage response pathways. This is the first major review of this family of proteins in more than 6 years. I have no criticisms of the subject or scope. The authors have incorporated much of what is currently known making this manuscript a useful reference. Minor point, there is a typo on line #290: cycline → cyclin.
Response: We thank the reviewer very much for recognizing the significance of our review. The typo has been corrected.